

# Development and validation of a nomogram for assessment postoperative sodium disturbance in PAs patients: a retrospective cohort study

Wenpeng Li[1,2,*], Dongfang Tang[3,*], Qiwei Wang[2], Shiwei Li[1], Wenbo Zhao[4] and Lili You[5]

[1] Department of Neurosurgery, Sun Yat-sen Memorial Hospital, Guang Zhou, China
[2] Orthopedics, Sun Yat-sen University, Guangzhou, China
[3] Department of Neurosurgery, Xiangyang Central Hospital, Affiliated Hospital of Hubei University of Arts and Science, Xiangyang, China
[4] Neurosurgery, Second Hospital of Shanxi Medical University, Shanxi, China
[5] Department of Endocrinology, Sun Yat-sen Memorial Hospital, Guang Zhou, China
[*] These authors contributed equally to this work.

Corresponding authors
Lili You, youlli@mail.sysu.edu.cn
Wenbo Zhao, 18234135012@163.com

## ABSTRACT

**Background**. Pituitary adenomas (PAs) are neuroendocrine tumors located in the sellar region. Surgery, being the primary treatment option for most PAs, is known to cause disruptions in sodium metabolism.

**Objective**. To develop and validate a nomogram for assessment the incidence of postoperative sodium disturbance (SD) in patients with PAs.

**Methods**. In this retrospective study, 208 patients with PAs who underwent resection surgery between 2013 and 2020 were included. Various demographic characteristics, clinical features and laboratory data were analyzed as potential predictors of postoperative sodium disturbance (SD). LASSO regression were used to identify independent preoperative variables associated with SD. Logistic regression was employed to calculate odds ratios (ORs) and 95% confidence intervals (CIs). A nomogram was constructed to visualize these results and evaluated using metrics such as the area under the curve (AUC) for discrimination, the Hosmer-Lemeshow test for calibration and decision curve for usefulness assessment.

**Results**. The incidence of SD was 44.23% (92 cases out of 208). Six preoperative factors, including sex, types of PAs, phosphocreatine kinase (CK), serum iron (Fe), free fatty acids (NEFA) and mean corpuscular volume (MCV), were identified for constructing a predictive nomogram. The nomogram showed high accuracy, with AUC values of 0.851 (95% CI [0.799–0.923]) and 0.771 (95% CI [0.681–0.861]) in the training and validation datasets, respectively. Calibration assessment and decision curve analysis confirmed its good agreement and clinical utility.

**Conclusion**. A practical and effective nomogram for predicting SD after PAs surgery is presented in this study.

# INTRODUCTION

Pituitary adenomas (PAs) are common neuroendocrine tumors located in the sellar region, accounting for 10% to 15% of all diagnosed intracranial neoplasms (*Yuen et al., 2019*; *Molitch, 2017*). One of the frequent complications following surgery is sodium disturbance (SD) (*Hensen et al., 1999*; *Kristof et al., 2009*). Several studies have elucidated the underlying mechanisms, including the cerebral salt wasting (CSW) syndrome, syndrome of inappropriate antidiuretic hormone secretion (SIADH) and diabetes insipidus (DI) (*Yuen et al., 2019*).

Sodium is plays a vital role as an extracellular cation, with its concentration regulated by renal excretion and the Na-K-ATPase pump (*Tisdall et al., 2006*). Sodium disorders, such as hyponatremia and hypernatremia, are common electrolyte imbalances observed after PA surgery (*Braun & Mahowald, 2017*). However, severe SD can manifest as fatigue, dizziness, thirst, and even coma or demyelination. Therefore, early identification of patients at risk of developing SD would be highly beneficial. Although a few studies have examined clinical factors associated with postoperative hyponatremia (*Hensen et al., 1999*; *Du et al., 2022*), there is currently no nomogram is available to assess an individual's experiencing of early postoperative SD based on clinical parameters and preoperative routine biochemical results.

Thus, the objective of this study is to develop and validate a nomogram model that incorporates preoperative clinical parameters to predict postoperative SD in patients with PAs. The nomogram model has the potential to provide insights into identifying individuals at high risk of SD after surgery and aid in implementing personalized care approaches.

# MATERIALS AND METHODS

## Study design and patients

This retrospective study analyzed the medical records of 208 patients with PAs who underwent resection surgery at Sun Yat-sen Memorial Hospital between 2013 and 2020. Inclusion criteria were: (1) histopathologically diagnosed PAs, (2) patients aged 18 years or older, (3) first-time pituitary surgery, and (4) availability of postoperative serum sodium levels. Exclusion criteria included use of drugs regulating serum sodium within 3 months before surgery, and enteritis within the previous 3 months. A total of 221 patients underwent surgery, and 13 were excluded due to missing postoperative serum sodium levels. The patient selection flowchart was shown in Fig. 1. This study was approved by the Sun Yat-sen Memorial Hospital (approval SYSEC-KY-KS-2020-118).

## Data collection

Demographic data, preoperative clinical information, and laboratory indices were collected from the medical records. Education levels were categorized as low-education or high-education groups. Marriage status was classified as "Marriage" or "Others" (including unmarried, divorced, and cohabiting). Preoperative clinical profiles, including drug use, radiotherapy, hypertension, and diabetes mellitus, were recorded. PAs were classified based on preoperative hormonal analysis.

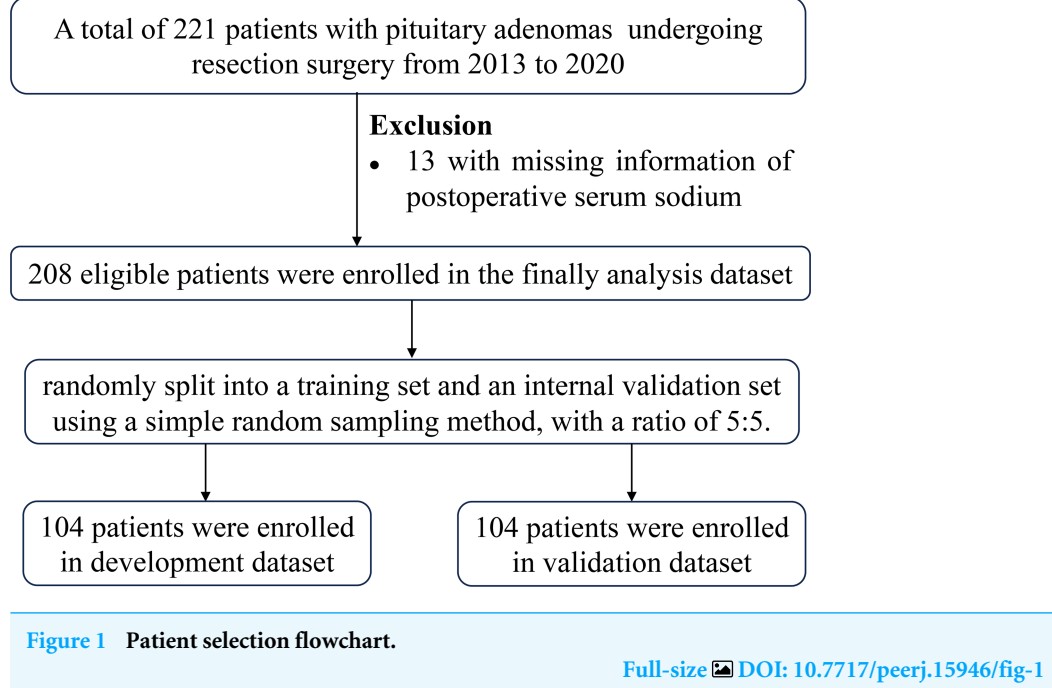

**Figure 1  Patient selection flowchart.**

## Evaluation of preoperative laboratory indices

Venous blood samples were obtained from individuals after an overnight fasting period of at least 10 h. All samples were collected prior to 9:00 am. Within 30 min of collection, the samples underwent centrifugation at 3,000 rpm for 10 min, and the measurements were conducted within 2 h. The Laboratory Department of Sun Yat-sen Memorial Hospital conducted all laboratory tests according to the manufacturers' instructions. The preoperative laboratory indicators included coagulation routine, blood routine examination and biochemical indicators (As shown in Table S1) and measured on Sysmex CS-5100 system TM (Siemens Healthcare Diagnostics, Erlangen, Germany), Sysmex XN-9000 analyzer (Sysmex Corporation, Kobe, Japan) and Beckman Coulter AU 5800 (Beckman Coulter Inc., Brea, CA, USA), respectively.

## Definition of postoperative SD

Serum sodium concentrations were measured immediately after surgery and on the second postoperative day using an automated analyzer. SD was defined as a serum sodium level of below 135 mmol/L or above 145 mmol/L on the day of surgery or the first day postoperative.

## Development of an individualized prediction model

Missing data were handled through multivariate normal imputation assuming "missing at random". The detailed information related to missing data for each variable was shown in Table S2. Variables with missing percentages exceeding 20%, such as iron saturation (70.19%), and serum ferritin (62.50%), creatinine and uric acid nitrogen (69.23%), cholylglycine (61.54%), apolipoprotein E (32.21%), retinol-binding protein (32.69%) and leucine aminopeptidase (21.63%), were excluded prior to multiple imputation. Multiple imputations ($m = 5$) were performed using the mice R package with chained equations.

Five complete datasets were obtained, and the first dataset was used for routine statistical analysis. Data sets with original and complete information did not differ significantly after multiple imputation.

A total of 208 PAs patients were randomly divided into a training set and an internal validation set using a 5:5 ratio. LASSO logistic regression was employed to select the best predictors for SD in the training set. Logistic regression analysis included variables such as age, gender, types of PAs, heart rate, total bilirubin (TBIL), carbon dioxide combining power (CO2-CP), creatine kinase (CK), serum iron (Fe), transferrin (Tf), adenosine deaminase (ADA), free fatty acids (NEFA), hemoglobin (HGB), mean corpuscular volume (MCV) and platelet crit (PCT). Akaike's stepwise selection was used for variable selection in multivariate logistic regression. A nomogram was constructed to visualize the risk factors, which was assessed for discrimination (area under the curve, AUC), calibration (Hosmer-Lemeshow test), and usefulness (decision curve).

### Statistical analyses

Continuous variables, skewed distribution variables, and categorical variables were tested using student's test, Mann–Whitney $U$ test, and chi-square test, respectively. Odds ratios (ORs) and 95% confidence intervals (CIs) were calculated using logistic regression analysis. Statistical analyses were performed using R (two-sided $P$ values $<0.05$ considered statistically significant).

## RESULTS

### The PAs patients' characteristics

The incidence of SD was 44.23% (92 cases/208 total number, hypernatremia = 50 and hyponatremia = 42). Ninety-two out of the 208 PAs patients developed SD. The prevalence of male was 50.96% (106/208) and the mean age was 50.57 ± 13.96 years for total population. Based on a random distribution of 208 PAs patients into the development and validation sets at the ratio of 5:5, the incidence of SD was 39.42% (41/104 cases) and 49.04% (51/104 cases), respectively (Table 1).

### Baseline biochemical characteristics of patients with and without SD

In Table 2, the baseline biochemical profiles of 208 participants were analyzed based on the presence or absence of prevalent SD. When compared to individuals with normal sodium levels, subjects with hyper-hypo-natremia showed significant elevations in prothrombin time (PT, S), alanine aminotransferase (ALT, U/L), direct bilirubin (DBIL, umol/L), phosphocreatine kinase (CK, U/L), lactate dehydrogenase (LDH, U/L), leucyl aminopeptidase (LAP, U/L), and free fatty acids (NEFA, umol/L). Conversely, hyper-hypo-natremia subjects exhibited lower levels of serum magnesium (Mg, mmol/L), carbon dioxide-binding capacity (CO2-CP, mmol/L), albumin/globulin ratio (A/G), serum iron (Fe, umol/L), and hemoglobin (HGB, g/L) ($P < 0.05$).

Li et al. (2023), *PeerJ*, DOI 10.7717/peerj.15946

**Table 1  Characteristics of pituitary patients in the development and validation sets.**

| Variables | Total | Training dataset ($N = 104$) | | | | Validation dataset ($N = 104$) | | | |
|---|---|---|---|---|---|---|---|---|---|
| | | Normal sodium ($N = 63$) | Hyper-hypo-natremia ($N = 41$) | $t/w/\chi^2$ | $P$ | Normal sodium ($N = 53$) | Hyper-hypo-natremia ($N = 51$) | $t/w/\chi^2$ | $P$ |
| Male, $n$ (%) | 106(50.96) | 36(57.14) | 20(48.78) | 0.403 | 0.526 | 26(49.06) | 24(47.06) | <0.001 | 0.994 |
| Age, years | 50.57 ± 13.96 | 50.34 ± 13.20 | 47.95 ± 14.45 | 0.853 | 0.396 | 52.37 ± 12.63 | 51.07 ± 15.75 | 0.463 | 0.645 |
| High school or above, $n$ (%) | 79(37.98) | 26(41.27) | 17(41.46) | <0.001 | 1.000 | 19(35.84) | 17(33.33) | 0.004 | 0.949 |
| Marriage, $n$ (%) | 178(85.58) | 53(84.13) | 36(87.80) | 2.810 | 0.245 | 49(92.45) | 40(78.43) | 7.208 | 0.066 |
| Hypertension, $n$ (%) | 76(36.54) | 22(34.92) | 15(36.59) | <0.001 | 1.000 | 19(35.85) | 20(39.22) | 0.023 | 0.879 |
| Diabetes, $n$ (%) | 45(21.63) | 12(19.05) | 10(24.39) | 0.165 | 0.685 | 13(24.53) | 10(19.61) | 0.136 | 0.713 |
| Nonfunctioning PA, $n$ (%) | 165(79.33) | 44(69.84) | 37(90.24) | 9.869 | 0.043 | 41(77.36) | 43(84.31) | 2.639 | 0.620 |
| Tumor diameter, cm | 2.40[1.23, 3.20] | 2.40[1.15, 2.90] | 2.70[1.60, 3.72] | 1060 | 0.124 | 2.20[1.53, 2.70] | 2.60[1.15, 3.55] | 1086.5 | 0.085 |
| Weight, kg | 65.40 ± 12.34 | 66.80 ± 11.14 | 65.63 ± 14.84 | 0.431 | 0.668 | 64.36 ± 12.01 | 64.58 ± 12.07 | −0.095 | 0.925 |
| BMI, kg/m$^2$ | 25.15 ± 5.22 | 25.03 ± 5.03 | 26.16 ± 6.04 | −0.963 | 0.339 | 29.97 ± 4.53 | 24.77 ± 5.53 | 0.197 | 0.844 |
| SBP, mmHg | 124.70 ± 19.94 | 123.95 ± 17.74 | 124.05 ± 21.77 | −0.024 | 0.981 | 125.53 ± 19.91 | 125.31 ± 21.49 | 0.053 | 0.958 |
| DBP, mmHg | 80.51 ± 12.92 | 81.13 ± 13.05 | 81.41 ± 12.93 | −0.110 | 0.912 | 78.42 ± 9.72 | 81.20 ± 15.55 | −1.09 | 0.279 |
| Heart rate, times/minutes | 79.18 ± 13.30 | 77.43 ± 13.67 | 81.88 ± 12.96 | −1.674 | 0.098 | 77.17 ± 11.02 | 81.22 ± 14.86 | −1.566 | 0.121 |

**Notes.**

Data were mean ±SD or median (IQR) for skewed variable or numbers (proportions) for categorical variables.

$P$ values were for the analysis of students' $t$ test, Wilcox test or $\chi^2$ analysis across the groups.

PA, pituitary adenomas; BMI, body mass index; SBP, systolic blood pressure; DBP, diastolic blood pressure.

**Table 2 Characteristics of pituitary patients' preoperative biochemistry indices.**

| Preoperative variables | Total | Normal sodium (N = 116) | Hyper-hypo-natremia (N = 92) | P |
|---|---|---|---|---|
| prothrombin time (PT, S) | 11.4 (1.13) | 11.3 (1.18) | 11.6 (1.04) | 0.041 |
| prothrombin activity (PTA, %) | 101 (19.9) | 103 (18.6) | 98.7 (21.4) | 0.176 |
| prothrombin time ration (PTR) | 1.04 (0.66) | 0.99 (0.07) | 1.11 (0.98) | 0.240 |
| prothrombin time international normalized ratio (PTINR) | 1.00 (0.08) | 0.99 (0.07) | 1.01 (0.09) | 0.094 |
| fibrinogen (g/L) | 3.16 (1.02) | 3.11 (0.99) | 3.23 (1.05) | 0.390 |
| activated partial thromboplastin time (APTT, S) | 26.9 (5.14) | 26.5 (4.44) | 27.4 (5.89) | 0.225 |
| thrombin time (TT, S) | 18.5 (1.43) | 18.4 (1.42) | 18.6 (1.45) | 0.261 |
| D-dimer (mg/L) | 0.56 (1.54) | 0.38 (0.65) | 0.79 (2.18) | 0.080 |
| alanine aminotransferase (ALT, U/L) | 29.2 (26.4) | 25.7 (20.2) | 33.5 (32.3) | 0.044 |
| aspartate aminotransferase (AST, U/L) | 28.5 (30.0) | 24.9 (16.0) | 33.0 (41.0) | 0.075 |
| total bilirubin (TBIL, umol/L) | 13.3 (6.50) | 12.5 (4.36) | 14.4 (8.38) | 0.056 |
| direct bilirubin (DBIL, umol/L) | 2.35 (1.44) | 2.15 (0.98) | 2.61 (1.83) | 0.034 |
| indirect bilirubin (IBIL, umol/L) | 12.9 (17.1) | 11.1 (11.7) | 15.1 (22.0) | 0.119 |
| $\Upsilon$-glutamyl transferases (GGT, U/L) | 36.5 (33.9) | 34.1 (34.1) | 39.6 (33.6) | 0.253 |
| alkaline phosphatase (ALP, U/L) | 81.0 (25.3) | 81.0 (24.3) | 81.0 (26.6) | 0.995 |
| serum potassium (K, mmol/L) | 3.97 (0.38) | 3.99 (0.40) | 3.96 (0.35) | 0.604 |
| serum sodium (Na, mmol/L) | 139 (9.76) | 140 (3.14) | 137 (14.1) | 0.053 |
| serum chlorine (CL, mmol/L) | 105 (4.28) | 105 (3.27) | 104 (5.25) | 0.112 |
| serum calcium (Ga, mmol/L) | 12.0 (140) | 2.30 (0.10) | 24.3 (211) | 0.320 |
| serum phosphorus (P, mmol/L) | 1.23 (0.32) | 1.23 (0.22) | 1.22 (0.42) | 0.841 |
| serum magnesium (Mg, mmol/L) | 0.87 (0.08) | 0.88 (0.08) | 0.86 (0.09) | 0.033 |
| cystatin C (CysC, mg/L) | 0.82 (0.23) | 0.82 (0.22) | 0.83 (0.25) | 0.785 |
| urea (mmol/L) | 5.07 (2.98) | 5.31 (3.32) | 4.76 (2.46) | 0.170 |
| creatinine (umol/L) | 79.4 (65.7) | 75.0 (19.5) | 84.9 (96.3) | 0.333 |
| capable of binding carbon dioxide (CO2-CP, mmol/L) | 24.8 (3.20) | 25.2 (2.73) | 24.2 (3.66) | 0.040 |
| uric acid (UA, umol/L) | 368 (138) | 364 (113) | 375 (165) | 0.591 |
| glucose (mmol/L) | 9.08 (36.9) | 9.20 (40.3) | 8.94 (32.4) | 0.958 |
| $\beta$-hydroxybutyric acid (HBUT, mmol/L) | 0.30 (0.94) | 0.21 (0.72) | 0.41 (1.15) | 0.163 |
| total cholesterol (T-Chol, mmol/L) | 5.24 (1.47) | 5.35 (1.36) | 5.11 (1.59) | 0.251 |
| triglycerides (TG, mmol/L) | 2.09 (1.43) | 2.05 (1.51) | 2.13 (1.33) | 0.673 |
| low-density lipoprotein-cholesterol (LDL-C, mmol/L) | 1.18 (0.34) | 1.19 (0.35) | 1.15 (0.33) | 0.368 |
| high-density lipoprotein-cholesterol (HDL-C, mmol/L) | 3.41 (1.05) | 3.47 (1.00) | 3.34 (1.11) | 0.401 |
| apolipoprotein A1 (ApoA1, g/L) | 1.21 (0.31) | 1.21 (0.24) | 1.21 (0.38) | 0.924 |
| apolipoprotein B (ApoB, g/L) | 1.02 (0.38) | 1.04 (0.43) | 0.99 (0.31) | 0.375 |
| prealbumin (PA, g/L) | 0.29 (0.11) | 0.30 (0.10) | 0.29 (0.12) | 0.565 |
| total protein (TP, g/L) | 69.6 (7.87) | 70.0 (5.83) | 69.2 (9.88) | 0.499 |
| albumin (ALB, g/L) | 40.8 (4.16) | 40.9 (4.02) | 40.6 (4.34) | 0.650 |
| globulin (GLB, g/L) | 29.3 (4.40) | 29.1 (4.73) | 29.5 (3.97) | 0.488 |

**Table 2** (*continued*)

| Preoperative variables | Total | Normal sodium (*N* = 116) | Hyper-hypo-natremia (*N* = 92) | *P* |
|---|---|---|---|---|
| albumin/globulin (A/G) | 1.41 (0.26) | 1.44 (0.30) | 1.36 (0.21) | 0.018 |
| total bile acid (TBA, umol/L) | 4.89 (5.38) | 4.73 (4.49) | 5.09 (6.35) | 0.647 |
| phosphocreatine kinase (CK, U/L) | 139 (151) | 118 (74.8) | 165 (209) | 0.040 |
| lactate dehydrogenase (LDH, U/L) | 207 (76.0) | 191 (60.6) | 226 (88.4) | 0.001 |
| creatine kinase lsoenzyme (CK-MB, U/L) | 14.4 (12.3) | 13.3 (5.14) | 15.9 (17.5) | 0.168 |
| C-reaction protein (CRP, mg/L) | 43.0 (509) | 68.2 (681) | 11.1 (29.2) | 0.369 |
| cholinesterase (CHE, U/L) | 7993 (2148) | 8160 (2091) | 7783 (2211) | 0.212 |
| leucyl aminopepidase (LAP, U/L) | 34.3 (9.88) | 33.0 (9.19) | 36.0 (10.5) | 0.032 |
| $\alpha$-L-fucosidase (AFU, U/L) | 27.4 (9.00) | 27.5 (7.64) | 27.2 (10.5) | 0.788 |
| lipase (LIP, U/L) | 45.5 (47.9) | 39.5 (19.7) | 53.2 (68.0) | 0.064 |
| serum amylase (SAMY, U/L) | 80.9 (87.9) | 73.6 (26.4) | 90.2 (129) | 0.227 |
| serum iron (Fe, umol/L) | 17.0 (9.75) | 18.7 (11.3) | 14.9 (6.89) | 0.003 |
| unsaturated iron binding capacity (UIBC, umol/L) | 38.5 (13.2) | 37.0 (11.8) | 40.3 (14.6) | 0.079 |
| total iron binding capacity (TIBC, umol/L) | 54.8 (13.6) | 54.7 (11.5) | 54.8 (15.9) | 0.968 |
| transferrin (Tf, g/L) | 2.41 (0.88) | 2.33 (0.63) | 2.51 (1.12) | 0.181 |
| adenosine dehydrogenase (ADA, U/L) | 11.2 (9.05) | 10.0 (3.32) | 12.7 (13.0) | 0.059 |
| superoxide dismutase (SOD, U/ml) | 151 (30.2) | 150 (18.5) | 151 (40.6) | 0.902 |
| free fatty acids (NEFA, umol/L) | 447 (235) | 397 (183) | 510 (276) | 0.001 |
| white blood cells (WBC, $10^9$/L) | 7.11 (2.64) | 6.80 (2.29) | 7.50 (3.00) | 0.067 |
| red blood cell (RBC, $10^{12}$/L) | 4.62 (0.77) | 4.64 (0.64) | 4.60 (0.90) | 0.719 |
| hemoglobin (HGB, g/L) | 131 (19.6) | 133 (16.5) | 127 (22.6) | 0.039 |
| platelet count (PLT, $10^9$/L) | 259 (65.7) | 257 (62.9) | 261 (69.3) | 0.652 |
| hematocrit (HCT) | 1.90 (21.7) | 0.40 (0.05) | 3.80 (32.7) | 0.322 |
| mean red cell volume (MCV, fL) | 90.3 (55.9) | 86.8 (10.0) | 94.6 (83.4) | 0.375 |
| mean corpuscular hemoglobin (MCH, pg) | 29.1 (5.27) | 28.9 (2.62) | 29.3 (7.38) | 0.679 |
| mean corpuscular hemoglobin concentration (MCHC, g/L) | 330 (25.5) | 330 (12.1) | 329 (35.9) | 0.650 |
| coefficient of variation of red blood cell distribution (RDW-CV) | 1.80 (23.9) | 0.13 (0.01) | 3.91 (35.9) | 0.319 |
| standard deviation of red blood cell distribution (RDW-SD, fL) | 42.1 (5.57) | 42.0 (3.98) | 42.2 (7.12) | 0.797 |
| lymphocyte percentage (LYM, %) | 32.2 (11.7) | 33.2 (11.0) | 30.9 (12.5) | 0.176 |
| neutrophil percentage (NEUT, %) | 58.5 (13.5) | 57.8 (12.1) | 59.5 (15.2) | 0.366 |
| percentage of monocyte (MONO, %) | 6.54 (5.37) | 5.90 (2.09) | 7.34 (7.68) | 0.083 |
| eosinophils percentage (EOS, %) | 2.58 (2.59) | 2.79 (2.69) | 2.33 (2.44) | 0.195 |
| basophils percentage (BASO, %) | 0.36 (0.34) | 0.37 (0.33) | 0.35 (0.36) | 0.559 |
| lymphocyte (LYM, $10^9$/L) | 2.07 (0.73) | 2.12 (0.70) | 2.02 (0.77) | 0.352 |
| neutrophils (NEUT, $10^9$/L) | 4.36 (2.46) | 4.08 (2.04) | 4.72 (2.87) | 0.070 |
| monocytes (MONO, $10^9$/L) | 0.47 (0.61) | 0.40 (0.21) | 0.56 (0.88) | 0.097 |
| eosinophils (EOS, $10^9$/L) | 0.17 (0.16) | 0.18 (0.16) | 0.15 (0.17) | 0.280 |
| basophils (BASO, $10^9$/L) | 0.02 (0.02) | 0.03 (0.02) | 0.02 (0.03) | 0.763 |

**Table 2** (*continued*)

| Preoperative variables | Total | Normal sodium (N = 116) | Hyper-hypo-natremia (N = 92) | P |
|---|---|---|---|---|
| plateletocrit (PCT, %) | 0.27 (0.07) | 0.26 (0.06) | 0.27 (0.08) | 0.285 |
| mean platelet volume (MPV, fL) | 10.3 (1.62) | 10.3 (1.39) | 10.3 (1.88) | 0.875 |
| platelet distribution width (PDW, fL) | 12.4 (7.47) | 12.7 (9.69) | 12.0 (2.82) | 0.468 |
| platelet large cell ratio (P-LCR, %) | 28.1 (9.19) | 27.8 (8.73) | 28.5 (9.76) | 0.599 |
| percentage of circulating reticulocytes (RET, %) | 1.64 (1.46) | 1.50 (0.56) | 1.82 (2.10) | 0.163 |
| reticulocyte (RET, $10^9$/L) | 68.0 (30.1) | 68.4 (25.5) | 67.5 (35.3) | 0.848 |
| immature reticulocyte fraction (IRF, %) | 10.5 (21.6) | 8.33 (5.31) | 13.2 (31.8) | 0.151 |

## LASSO regression feature selection and restricted cubic splines analysis

The LASSO regression was used to identified candidates for predicting SD (Figs. 2A and 2B), and RCS analysis were used to evaluate the correlated indices (including TBIL, CO2-CP, CK, Fe, Tf, ADA, NEFA, HGB, MCV, PCT, as shown in Figs. 3A to 3J, respectively) with the risk of postoperative SD, and there was no remarkable nonlinear association between the 10 preoperative laboratory indices and risk of SD ($P > 0.05$, excluding Fe and PCT for $P$ value was 0.041 and 0.026, respectively). Univariate and multivariate logistic regression analyses were utilized to examine the associations of the basic clinical features and ten selected preoperative laboratory testing indices, including age, gender, types of PAs, heart rate, TBIL, CO2-CP, CK, Fe, Tf, ADA, NEFA, HGB, MCV and PCT with prevalence of SD. The ten selected indices were categorized into tertiles. Multivariate logistic regression with Akaike's stepwise selection used to identify independent factors. In multivariate analysis, sex, type of PA, CK, Fe, NEFA, and MCV showed significant associations with prevalence of SD. High levels of CK and MCV were associated with increased risk, while higher levels of Fe and NEFA were associated with decreased risk ($P < 0.05$). As shown in Table 3.

## Nomogram model for postoperative SD and clinical use

LASSO regression and logistic regression identified sex, types of PAs, CK, Fe, NEFA and MCV as predictors for incidence of postoperative SD in nomogram (Fig. 4). ROC was used to assess the ability of nomogram in the development dataset (Fig. 5A) and training dataset (Fig. 5B). The area under curves (AUCs) of ROC were 0.851 (training dataset, 95% CI [779–0.923]) and 0.771 (validation dataset, 95% CI [0.681–0.861]), respectively. Both training ($P = 0.393$) and validation ($P = 0.179$) sets showed good agreement on the calibration curve (Figs. 6A and 6B). DCA was used to examine the nomogram usefulness (Fig. 7). In both the development and validation datasets, the net effect was positive within the threshold probability was between 0.2 and 0.9.

## DISCUSSION

By selecting conventional demographic characteristics and preoperative laboratory indicators, we aimed to identify the risk factors, and developed a user-friendly prediction tool for early postoperative SD in PAs. Six preoperative features, including sex, types of

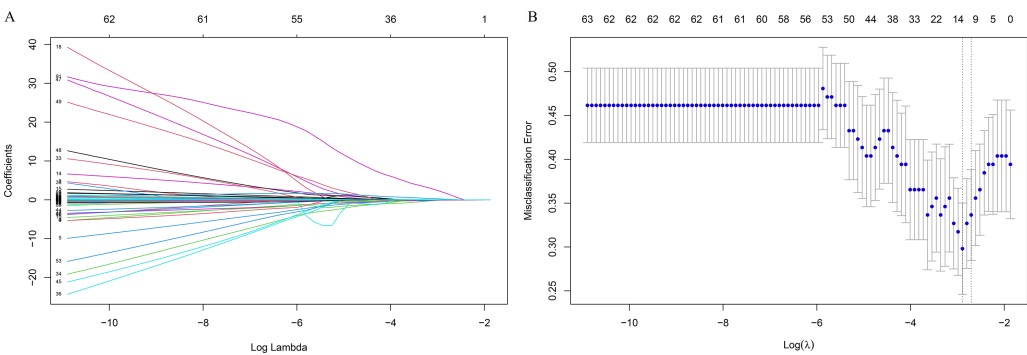

**Figure 2** **Feature selection using the least absolute shrinkage and selection operator (LASSO) binary logistic regression model.** (A) Model LASSO. Color lines represent the preoperative indices associated with postoperative sodium disturbance. The *x*-axis represents the alpha (cutoff), and the *y*-axis represents the shrink effect value. LASSO coefficient profiles of the 93 texture features. A coefficient profile plot was produced against the log ($\lambda$) sequence. A vertical line was drawn at the value selected using 10-fold cross-validation, where optimal $\lambda$ resulted in 10 nonzero coefficients. (B) Tuning parameter (lambda) selection in the LASSO model used 10-fold cross-validation *via* minimum criteria for risk of postoperative sodium disturbance. The area under the receiver operating characteristic (AUC) curve was plotted *versus* log($\lambda$). Dotted vertical lines were drawn at the optimal values by using the minimum criteria.

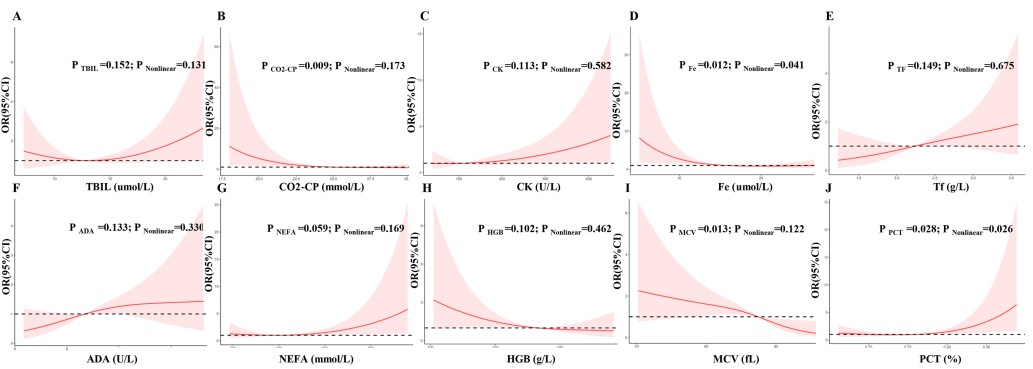

**Figure 3** **Multivariable logistic regression models restricted cubic spline analysis the selected variables by LASSO regression analyses on a continuous scale and risk for incidence postoperative sodium disturbance.** Odds ratios are represented by a solid line, and the 95% confidence intervals are represented by the pink area. (A) to (J) were restricted cubic spline modeling of the association of TBIL, CO2-CP, CK, Fe, Tf, ADA, NEFA, HGB, MCV and PCT with risk of postoperative sodium disturbance.

PAs, CK, Fe, NEFA, and MCV, were identified for construction of the nomogram model. One of the highlights of this model that almost all predictors were derived from routine preoperative biochemistry indices, making it easy and convenient for junior doctors with limited clinical experience to utilize in clinical practice. This nomogram can help identify high-risk patients, closely monitor postoperative serum sodium, and intervene as early as possible, thus saving medical resources and minimizing the risk of adverse reactions.
**Table 3 Risk factors for postoperative hyper-hypo-natremia in training dataset ($n = 104$).**

| Variables | Groups | Univariate model | | Multivariate model | |
|---|---|---|---|---|---|
| | | OR (95%CI) | P trend | OR (95%CI) | P trend |
| Age, years | [18.7,50] | 1.00 | – | | |
| | (50,76.4) | 0.67(0.30–1.47) | 0.317 | | |
| Sex | Male | 1.00 | – | 1.00 | – |
| | Female | 1.40(0.64, 3.10) | 0.404 | 4.55(1.40–16.80) | 0.015 |
| Type of PA | Non-functioning PA | 1.00 | – | 1.00 | – |
| | Functioning PA | 0.25(0.07–0.74) | 0.020 | 0.36(0.14–0.69) | 0.010 |
| Heart rate, times/minutes | [49,73] | 1.00 | – | | |
| | (73,83) | 1.39(0.52–3.75) | 0.511 | | |
| | (83,114) | 2.05(0.77–5.63) | 0.153 | | |
| TBIL, umol/L | [4.8,10.7] | 1.00 | – | | |
| | (10.7,14.8) | 0.84(0.32–2.21) | 0.723 | | |
| | (14.8,50.8) | 1.43(0.54–3.84) | 0.478 | | |
| CO2-CP, mmol/L | [15,23] | 1.00 | – | | |
| | (23,25) | 0.30(0.11–0.78) | 0.016 | | |
| | (25,32) | 0.25(0.09–0.65) | 0.006 | | |
| CK, U/L | [23,85] | 1.00 | – | 1.00 | – |
| | (85,133) | 1.92(0.72–5.28) | 0.196 | 4.85(1.25–21.93) | 0.029 |
| | (133,1.08e+03) | 1.89(0.70–5.28) | 0.211 | 6.31(1.46–32.20) | 0.018 |
| Fe, umol/L | [3.2,12.8] | 1.00 | – | 1.00 | – |
| | (12.8,18.8) | 0.15(0.05–0.42) | <0.001 | 0.08(0.02–0.30) | 0.001 |
| | (18.8,35.8) | 0.32(0.12–0.85) | 0.024 | 0.23(0.06–0.82) | 0.029 |
| Tf, g/L | [1.02,2] | 1.00 | – | | |
| | (2,2.46) | 1.48(0.54–4.11) | 0.446 | | |
| | (2.46,4.81) | 2.81(1.06–7.84) | 0.042 | | |
| ADA, U/L | [3.9,8.5] | 1.00 | – | | |
| | (8.5,11.2) | 1.45(0.55–3.95) | 0.455 | | |
| | (11.2,20.1) | 1.94(0.73–5.28) | 0.186 | | |
| NEFA, mmol/L | [98,314] | 1.00 | – | 1.00 | – |
| | (314,504) | 0.41(0.14–1.14) | 0.092 | 0.21(0.05–0.85) | 0.035 |
| | (504,1.3e+03) | 1.81(0.70–4.81) | 0.227 | 2.37(0.65–9.22) | 0.198 |
| HGB, g/L | [4.79,124] | 1.00 | – | | |
| | (124,139) | 0.42(0.15–1.08) | 0.076 | | |
| | (139,180) | 0.54(0.20–1.41) | 0.213 | | |
| MCV, fL | [0.42,85.8] | 1.00 | – | 1.00 | – |
| | (85.8,90.1) | 0.61(0.23–1.55) | 0.297 | 0.32(0.08–1.17) | 0.097 |
| | (90.1,102) | 0.28(0.10–0.79) | 0.018 | 0.16(0.03–0.61) | 0.011 |
| PCT, % | [0.01,0.23] | 1.00 | – | | |
| | (0.23,0.29) | 0.78(0.29–2.08) | 0.617 | | |
| | (0.29,0.49) | 2.01(0.76–5.40) | 0.161 | | |

**Notes.**

LASSO logistic regression was applied to select the best predictors for SD in the training set. Logistic regression analysis performed with: Age, gender, types of PAs, heart rate, total bilirubin (TBIL), carbon dioxide combining power (CO2-CP), creatine kinase (CK), Serum iron (Fe), transferrin (Tf), adenosine deaminase (ADA), free fatty acids (NEFA), hemoglobin (HGB), Mean corpuscular volume (MCV) and platelet crit (PCT). Akaike's stepwise selection was applied as variables selection in multivariate logistic regression.

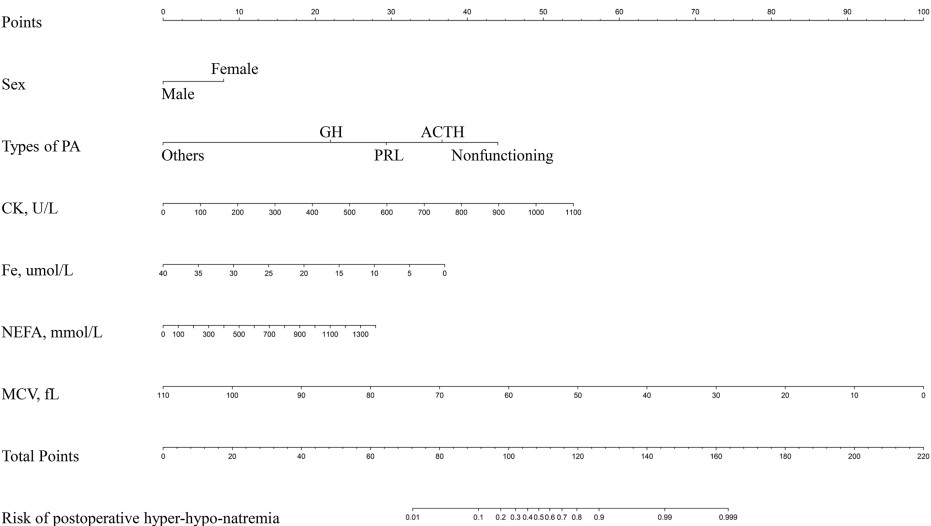

**Figure 4  Developed and validated preoperative indices nomogram.** The preoperative indices nomogram was developed with sex, type of pituitary adenoma, CK, Fe, NEFA and MCV.

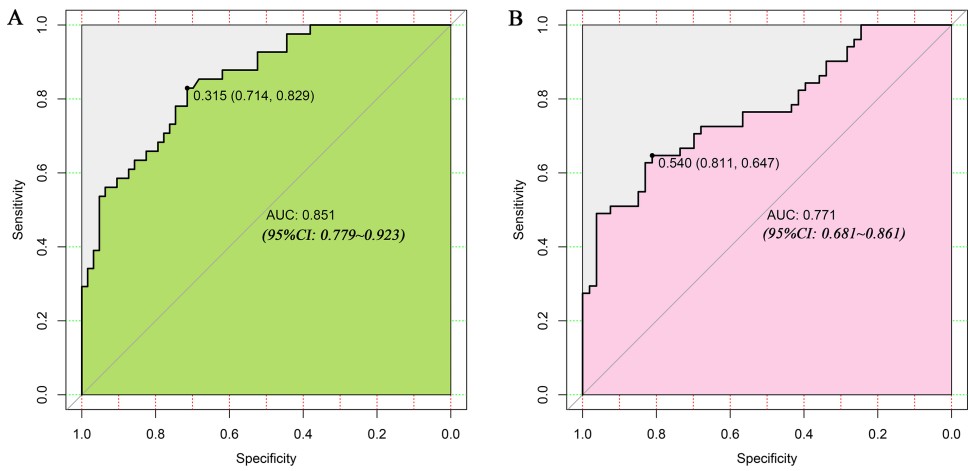

**Figure 5  Receiver operating characteristic curves (ROC) for the preoperative index's prediction model predicting postoperative sodium disturbance in training and validation sets.** AUC, area under the curve.

The preoperative pituitary function had been widely reported to adversely affect the rates of SD (*Yuen et al., 2019*). *Hensen et al. (1999)* reported that patients with ACTH-PAs had the highest risk for the development of delayed hyponatremia, which may be due to the antidiuretic hormone inhibitory properties of cortisol (*Berghorn et al., 1995*). In line with these findings, our study found that ACTH-PA was a risk factor for SD, and nonfunctioning PA was associated with a higher risk of postoperative SD. These findings can be explained in two ways. First, in neuroradiological series, the majority of macroadenomas are nonfunctioning PAs due to their nonspecific symptom (*Imran et al., 2016*; *Arita et al., 2006*). Several studies have reported that larger tumors are associated with
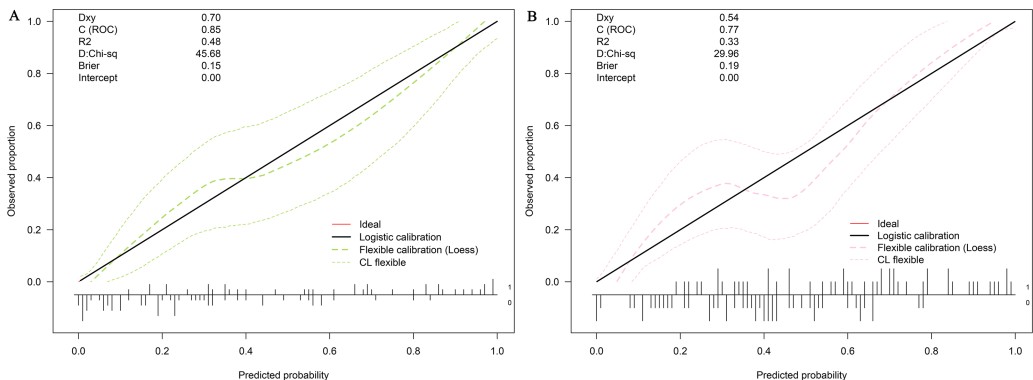

**Figure 6** Calibration curves of the nomogram for the incidence of postoperative sodium disturbance. (A) Development datasets; (B) Internal validation dataset. The *y*-axis represents the actual postoperative sodium disturbance rate. The *x*-axis represents the predicted postoperative sodium disturbance risk. The diagonal dotted represents a perfect prediction by an ideal model. The dotted line represents the performance of the nomogram, of which a closer fit to the diagonal dotted line represents a better prediction.

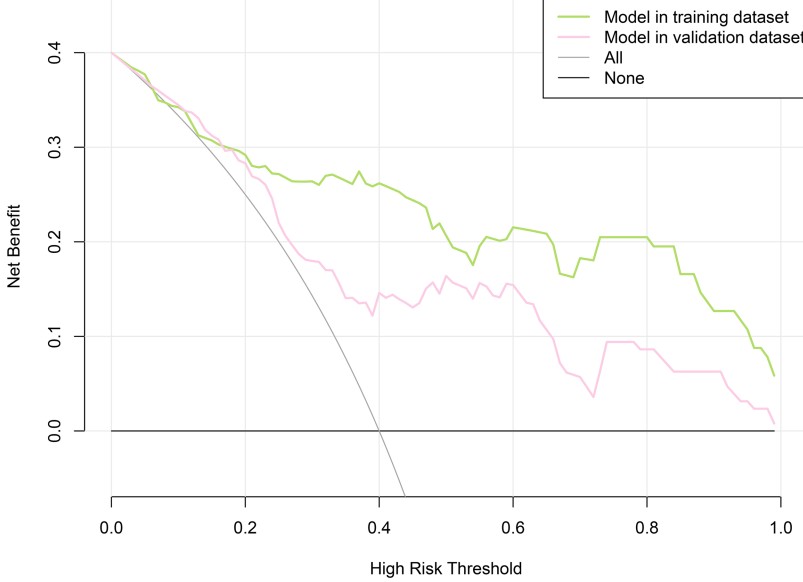

**Figure 7** Decision curve analysis for the nomogram model of postoperative sodium disturbance in development and internal validation datasets. The *y*-axis measures the net benefit. The *x*-axis indicates the threshold probability of the risk of postoperative sodium disturbance. The green line represents the nomogram in development set, and the red one represents the nomogram in internal validation set. The grey line represents the assumption that all patients have postoperative sodium disturbance. Thin black line represents the assumption that no patient has postoperative sodium disturbance.

a higher rate of DI (*Nomikos et al., 2004*; *Olson et al., 1997*) and significantly increased risk of sodium disturbance due to more extensive exploration of the sellar region (*Kelly, Laws Jr & Fossett, 1995*; *Staiger et al., 2013*). For instance, *Kelly, Laws Jr & Fossett (1995)* found that patients with large adenomas were more likely to develop late-onset hyponatremia

compared those with small tumors. Second, our study collected and evaluated serum sodium on the immediate postoperative day and/or postoperative day 1, while most studies primarily focus on the occurrence of delayed hyponatremia in the postoperative period (*Cote et al., 2016*).

Lower levels of Fe was closely correlated with an increased prevalence of postoperative SD. This association can be explained by the fact that lower Fe levels stimulates Na-K-ATPase (*Yin et al., 2003*), which drives potassium into and sodium out of the cell (*Feldmann et al., 2007*). Our study also revealed significant associations between postoperative SD and levels of CK, NEDA, and MVC. Previous clinical and laboratory studies have demonstrated that high CK levels affect sodium reabsorption, (*Brewster, 2018*) as Na-K-ATPase is tightly bound to CK and contributes to the rapid regeneration of ATP for driving sodium retention in the kidneys (*Ikeda, 1988*; *Brewster et al., 2018*). In a study involving 60 healthy men, it was observed that urinary sodium excretion decreased by 98.4 mmol/24 h per unit increase in log CK after adjusting for age and African ancestry (*Brewster et al., 2018*). Furthermore, our study found that higher levels of NEDA indicated an increased risk of postoperative SD. Previous studies have reported that NEFA inhibits Na-K-ATPase (*Iannello, Milazzo & Belfiore, 2007*; *Swarts et al., 1988*), suggesting its potential involvement in the regulation of SD. Additionally, levels of MVC, reflecting the status of red blood cell (RBC), were significantly correlated with postoperative SD in our study. The activity of Na-K-ATPase plays a role in controlling the surface area-to-volume ratio and cytoplasmic rheology, which are two major factors in erythrocyte deformability and clearance (*Lew & Tiffert, 2017*; *Maxwell et al., 2021*). Thus, low levels of MVC may reflect a phenomenon of decreased Na-K-ATPase function.

Our nomograms serve a dual purpose: they not only display the relevant predictors identified through multifactor regression analyses but also provide a simple graphical representation for predicting postoperative SD probabilities. All indicators are derived from demographics and routine preoperative biochemical indices, making the prediction process simpler and more convenient. This nomogram demonstrated excellent discriminative power, net benefit and good agreement between the predicted and the observed probability, as verified through ROC, decision curve, and calibration curve analyses. By effectively identifying high risk patients, preventative measures can be implemented in the perioperative setting to mitigate the incidence of SD.

However, there are several potential drawbacks. Firstly, this study conducted at a single center and was retrospective in nature, introducing the possibility of selection bias. Therefore, further validation through randomized controlled prospective multi-center studies is necessary before our model can be widely applied. Secondly, determining the exact duration of SD in each patient is challenging in a retrospective study, and thus we were unable to obtain information on the duration of SD in PAs patients. Thirdly, since the clinical characteristics and treatment methods for hypernatremia and hyponatremia are distinct, it would be more advantageous to develop separate prediction models for high sodium and low sodium conditions. However, due to the limited number of cases of hypernatremia ($n = 50$) and hyponatremia ($n = 42$) in this study, the small sample size became a main limitation in constructing such a nomogram. Lastly, patients received

surgery for a quite long period (2013–2020), but we did not include the year of surgery as a variable in our univariate and multivariate analyses.

## CONCLUSION

In conclusion, this study introduces a valuable and user-friendly nomogram for the individual prediction of postoperative SD in patients with PAs. By utilizing sex and routine preoperative laboratory indices, this nomogram provides a simple and accessible tool that can aid even less experienced clinicians and nurses in identifying individuals who require close monitoring for postoperative SD, thereby helping to prevent potential catastrophic complications.

## ACKNOWLEDGEMENTS

We are indebted to the participants of the present study for their persistent and outstanding support and to our colleagues for their valuable assistance.

### Funding

This work was supported by the Guangzhou Key Research and Development Program (No. 2023B03J0079) and Natural Science Foundation of Guangdong Province (2018A0303130193 and 2022A1515012393). The funders had no role in study design, data collection and analysis, decision to publish, or preparation of the manuscript.

### Grant Disclosures

The following grant information was disclosed by the authors:
Guangzhou Key Research and Development Program: 2023B03J0079.
Natural Science Foundation of Guangdong Province: 2018A0303130193, 2022A1515012393.

### Competing Interests

The authors declare there are no competing interests.

### Author Contributions

- Wenpeng Li conceived and designed the experiments, performed the experiments, analyzed the data, prepared figures and/or tables, authored or reviewed drafts of the article, and approved the final draft.
- Dongfang Tang conceived and designed the experiments, performed the experiments, authored or reviewed drafts of the article, and approved the final draft.
- Qiwei Wang performed the experiments, prepared figures and/or tables, and approved the final draft.
- Shiwei Li performed the experiments, prepared figures and/or tables, and approved the final draft.

- Wenbo Zhao conceived and designed the experiments, performed the experiments, authored or reviewed drafts of the article, and approved the final draft.
- Lili You conceived and designed the experiments, analyzed the data, prepared figures and/or tables, authored or reviewed drafts of the article, and approved the final draft.

### Human Ethics

The following information was supplied relating to ethical approvals (i.e., approving body and any reference numbers):

Sun Yat-sen Memorial Hospital (approval SYSEC-KY-KS-2020-118).

### Data Availability

The raw measurements are available in the Supplementary Files.

### Supplemental Information

Supplemental information for this article can be found online at http://dx.doi.org/10.7717/peerj.15946#supplemental-information.

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
