# Peer review of "Development and validation of a nomogram for assessment postoperative sodium disturbance in PAs patients: a retrospective cohort study"

_PeerJ, doi:10.7717/peerj.15946_

## Round 0.1 · original submission · Major Revisions

Please revise the manuscript as suggested.

Reviewer 1 ·

Basic reporting

the article fails to meet our standards

Experimental design

the article fails to meet our standards

Validity of the findings

the article fails to meet our standards

Additional comments

the article fails to meet our standards

Reviewer 2 ·

Basic reporting

Pituitary adenomas (PAs) are the neuroendocrine neoplasm and located in the sellar region. The surgery, as a choice for most PAs, is well known to cause sodium metabolism disturbances. In this study, the authors retrospectively included 208 patients with PAs who underwent resection surgery, and investigated the risk factors of postoperative sodium metabolism disturbances. The patients were randomly divided into the training cohort and the validation cohort, and the authors used LASSO, nomogram, and several other tools to identify and validate the risk factor. Overall, this manuscript is well structured and of novelty.

Experimental design

The variables including age, types of PAs, phosphocreatine kinase (CK), Serum iron (Fe), transferring (Tf), free fatty acids (NEFA) and hemoglobin (HGB), mean corpuscular volume (MCV), should better be considered as categorical variables but not continuous variables. For example, the increased risk from 30 to 40 years old, may not be the same as it from 70 to 80 years old.

The authors should better give more details on how they detect the CK, Fe, and so on. And they should better list the reference values of these variables, and compare the variables between patients with or without postoperative sodium metabolism disturbances.

Validity of the findings

no comment

Additional comments

no comment

Reviewer 3 ·

Basic reporting

none

Experimental design

none

Validity of the findings

none

Additional comments

Pituitary adenomas ( PAs ) are the neuroendocrine neoplasm and are located in the sellar region. The surgery, as a choice for most PAs, is well known to cause sodium metabolism disturbances. This study retrospectively investigated 208 patients with PAs who underwent resection surgery from 2013 to 2020, and analyzed a range of demographic characteristics, clinical features, and laboratory data. LASSO regression was used and a nomogram was established with a high AUCboth in training (AUC: 0.884, 95% CI: 0.815-0.952) and validation datasets (AUC:0.783, 95% CI: 0.695-0.870).

(1) There are a lot of grammatical errors in this manuscript, the authors should better carefully revise them. I recommend the authors use Grammarly (www.grammarly.com) to do this job.

(2) A flow chart is needed to clarify how the authors included and excluded patients in this analysis.

(3) Why were several variables with p > 0.05 (eg. the p-value of age is 0.383) in the univariate analysis still included in the multivariate analysis, as Table 2 shows?

(4) Since patients received surgery for a quite long period (2013-2020), I recommend the authors put the year at surgery in the univariate and multivariate analyses.

---

## Round 0.2 · accepted · Accept

This manuscript can be accepted now.

Reviewer 2 ·

Basic reporting

no comment

Experimental design

no comment

Validity of the findings

no comment

Additional comments

no comment

Reviewer 3 ·

Basic reporting

no comment

Experimental design

no comment

Validity of the findings

no comment

Additional comments

no comment